# High-risk human papillomavirus prevalence among South African women diagnosed with other STIs and BV

Wenkosi Qulu[1,2☯], Andile Mtshali[1,2☯], Farzana Osman[1], Nonsikelelo Ndlela[1,2], Lungelo Ntuli[1,2], Gugulethu Mzobe[1,2], Nivashnee Naicker[1], Nigel Garrett[1,3], Anne Rompalo[4], Adrian Mindel[1], Sinaye Ngcapu[1,2], Lenine Liebenberg[1,2,5]*

1 Centre for the AIDS Programme of Research in South Africa (CAPRISA), Durban, South Africa, 2 School of Laboratory Medicine and Medical Sciences, University of KwaZulu-Natal, Durban, South Africa, 3 School of Nursing and Public Health, Discipline of Public Health Medicine, University of KwaZulu-Natal, Durban, South Africa, 4 Johns Hopkins School of Medicine, Baltimore, Maryland, United States of America, 5 Centre for Epidemic Response and Innovation, Stellenbosch University, Cape Town, South Africa

☯ These authors contributed equally to this work.
* Liebenbergl@sun.ac.za

**Data Availability Statement:** All relevant data are within the paper and its Supporting Information files. Data are available by request on a dedicated

## Abstract

### Introduction

Human papillomavirus (HPV) infection is a leading cause of cervical cancer. Although this relies on infection and persistence of HPV in epithelial cells, often occurring in the context of other sexually transmitted infections (STIs) and bacterial vaginosis (BV), data on the relationships between these and their relative effects on epithelial barrier integrity in women remain sparse. This study describes the epidemiology of HPV combined with STI and/or BV prevalence and the relative impact on matrix metalloproteinases (MMPs) among South African women.

### Methods

Roche Linear Array was used for HPV genotyping in menstrual cup pellets of 243 HIV-negative women participating in the CAPRISA 083 cohort study. Vulvovaginal swabs were tested for *Chlamydia trachomatis*, *Neisseria gonorrhoeae*, and *Trichomonas vaginalis* using Xpert® CT/NG assay and lateral flow assay, and Gram staining was performed to diagnose BV using Nugent scoring criteria. Concentrations of 5 MMPs were measured in menstrual cup supernatants by multiplexed ELISA. Fisher's exact tests, Mann-Whitney U tests, and multivariable regression models determined associations between HPV infection, STI and/ or BV, and MMP concentrations.

### Results

HPV was prevalent in 34% of women (83/243; median 23 years, interquartile range (IQR) 21–27 years). Low-risk (lr) (71%, 59/83) and high-risk (hr)-HPV infections (54.2%, 45/83) were common. Hr-HPV was frequently detected in STI and/or BV-positive women compared to women without STIs or BV (p = 0.029). In multivariable analysis, BV was associated with

portal on the CAPRISA website (https://www.caprisa.org/Pages/CAPRISAStudies).

**Funding:** The study was funded by the Poliomyelitis Research Foundation (PRF grant number: 19/108) and the DST-NRF Center of Excellence in HIV Prevention, which is supported by the Department of Science and Innovation, and National Research Foundation (NRF). The CAPRISA 083 cohort study was funded by a United States – South African Program for Collaborative Biomedical Research grant through the South African Medical Research Council and the National Institute of Health (AI116759). WPQ received support from the CAPRISA Research Administration and Management Training Program (Grant # G11 TW010555-01). WPQ was funded by DST-NRF CoE in HIV Prevention (grant 96354). AM is funded by DST/NRF Innovation Postdoctoral Fellowship grant number: PDG210309589501. GM is funded by DST/NRF Innovation Postdoctoral Fellowship grant number: SFP180507326699. Poliomyelitis Research Foundation (PRF grant number: 18/16). The funders had no role in study design, data collection and analysis, decision to publish, or preparation of the manuscript.

**Competing interests:** The authors have declared that no competing interests exist.

increased odds of hr-HPV detection (OR: 2.64, 95%CI: 1.02–6.87, p = 0.046). Furthermore, Gardasil®9 vaccine-type strains were more frequently detected in women diagnosed with STI and/or BV (55.2%, 32/58 vs 24%, 6/25; p = 0.009). Among STI and/or BV-positive women, HPV detection was significantly associated with increased MMP-10 concentrations (b = 0.55, 95% CI 0.79–1.01; p = 0.022).

## Conclusion

Most women with hr-HPV had another STI and/or BV, emphasizing an urgent need for STI and BV screening and intensive scale-up of cervical cancer screening and HPV vaccination programmes. Furthermore, the study highlights the need for more extensive research to confirm and understand the relationship between HPV infection and barrier integrity.

## Introduction

Human papillomavirus (HPV), the most prevalent sexually transmitted virus worldwide, is the main etiologic agent for the development of cervical cancer [1, 2]. Although there are geographical disparities in HPV prevalence, sub-Saharan Africa has the highest global prevalence of HPV (ranging from 8.5 to 74.6%) among women with normal cervical cytology [3–5]. The risk for HPV infection is skewed toward young women who engage in sexual intercourse early in their reproductive years [6, 7]. In South Africa, young women <25 years of age are disproportionately affected by HPV and high-risk (hr)-HPV, with prevalence reaching up to 70% and 50%, respectively [8, 9]. The prevalence of multiple HPV infections and incidence of cervical cancer in young women varies widely depending on the different ethnic groups and geographic areas [10–12].

Young women are biologically susceptible to sexually transmitted infections (STIs), including HIV infection [13]. Studies have demonstrated that women living with HIV are 6 times more likely to develop cervical cancer compared to women without HIV [14]. STIs such as *Chlamydia trachomatis*, *Neisseria gonorrhoeae*, and *Trichomonas vaginalis* have been associated with HPV infection and persistence, leading to delayed infection clearance and possible induction of precancerous lesions or cervical intraepithelial neoplasia (CIN) type 1, 2, or 3 [15, 16]. Furthermore, bacterial vaginosis (BV), the most common cause of abnormal vaginal discharge among women of childbearing age, is associated with incident HPV infection and HPV persistence [17, 18]. The high prevalence of BV and multiple infections with sexually transmitted pathogens (such as HPV and other STIs) has been associated with increased HIV acquisition risk in sub-Saharan African women [19–21].

Although cervical cancer is preventable through HPV vaccination if it is administered before exposure, HPV vaccination coverage has been low while incidence rates remain high in resource-limited settings [19]. Data from South Africa have shown that approximately 3.2% of women in the general population harbour cervical HPV-16 or HPV-18 infection and 64.2% of invasive cervical cancers have been attributed to these genotypes [22]. In addition, HPV-35 and HPV-45 associated cervical precancers are more common in women of African descent and account for nearly 10% of cervical cancer cases [1, 23–25]. While the existing highly effective HPV vaccines have decreased HPV-16 prevalence, these do not target HPV-35 and may explain the increased prevalence observed in sub-Saharan African women [12, 25–27].

An intact squamous epithelium effectively protects against pathogen entry [28]. As HPV infects and replicates in the epithelium [29] cells might lose their epithelial characteristics and

develop a mesenchymal phenotype [30]. However the contribution of HPV infection to this transformation has not yet been fully elucidated. Matrix metalloproteinases (MMPs) are bio-markers of epithelial barrier integrity that play a critical role in the degradation of extracellular matrix, including remodelling of the tumour microenvironment, cell invasion, and development of metastatic disease [31–35]. In addition, MMPs are involved as essential molecules in multiple and diverse physiological processes of cancer, such as cell migration, differentiation, proliferation, apoptosis, inflammatory reactions, angiogenesis, and platelet aggregation [36]. While there is a relationship between MMPs and cervical cancer, it remains unknown whether HPV infection induces MMP expression in the female genital tract.

In this study, we determined the prevalence of HPV in a population of 243 young South African women attending primary healthcare in Durban, South Africa. We identified the most common circulating oncogenic genotypes and determined associations of BV and other STIs with HPV prevalence to test the hypothesis that HPV and concurrent STI and/or BV weaken the epithelial barrier integrity in the genital mucosa.

## Materials and methods

### Study design and population

This was a cross-sectional analysis of 243 young South African women who participated in the CAPRISA 083 cohort study between May 2016 and January 2017. All women provided written informed consent for participation in the study, and participants were only included in this study if consenting to specimen storage for assessment of the study's secondary objectives and any future research. CAPRISA 083 evaluated a combination of point-of-care (POC) STI testing, immediate treatment, and expedited partner therapy in non-pregnant, HIV-negative women, 18 years and older, who attended for sexual and reproductive care at a public health-care clinic in eThekwini, South Africa [37, 38]. Gram stain (Nugent score with 0–3 considered BV negative, 4–6 intermediate BV, and 7–10 BV positive) was used for BV screening [39]. STI testing included screening for *C. trachomatis*, *N. gonorrhoeae* by Xpert® CT/NG (Cepheid, Sunnydale, California, US) and for *Trichomonas vaginalis* (TV) by the OSOM® Rapid Tricho-monas Test (Sekisui Diagnostics, Lexington, MA, US). Women with confirmed STIs and/or BV were treated appropriately as per international guidelines. Any data that could potentially identify participants' information were anonymised throughout the research process. The authors adhered to the relevant data protection regulations and ethical guidelines to minimize the risk of unintentional disclosure of participants' identities. The study was approved by the Biomedical Research Ethics Committee of the University of KwaZulu-Natal (BE303/17).

### Measurement of matrix metalloproteinases in SoftCup supernatants

Concentrations of 5 MMPs (MMP-1, -2, -7, -9, and -10; MMP Panel-2 kit, Merck-Millipore, Missouri, U.S.A.) were measured in SoftCup supernatants collected at enrollment were measured on a Bio-Plex 200 Array Reader system (Bio-Rad Laboratories Inc®, Hercules, California). Assays were conducted according to the manufacturer's instructions. The sensitivity of the kits ranged between 0.2 and 45.2 pg/ml for each cytokine measured and between 2 and 200 pg/ml for each MMP. Bio-Plex manager software (version 5.0; Bio-Rad Laboratories Inc®) was used to analyze the data and all analyte concentrations were extrapolated from the standard curves using a 5-parameter logistic (PL) regression equation. Analyte concentrations that were below the lower limit of quantification were reported as the mid-point between zero and the lowest concentration measured for each analyte. For analyte readings above the upper limit of detection, concentrations were reported as halfway between the highest concentration and the upper limit of the standard curve.

## HPV DNA detection and genotyping in cervicovaginal specimens

HPV DNA was extracted in matching menstrual cup pellet specimen using an automated MagNA pure instrument (Roche Diagnostics, Indianapolis, IN, USA) and amplified using the Roche Linear Array® HPV Genotyping Test kit (Roche Diagnostics, Indianapolis, IN, USA) according to the manufacturer instructions. The assay identifies 37 HPV genotypes [HPV-6, -11, -16, -18, -26, -31, -33, -35, -39, -40, -42, -45, -51, -52, -53, -54, -55, -56, -58, -59, -61, -62, -64, -66, -67, -68, -69, -70, -71, -72, -73, -81, -82, -83, -84, -89 (HPV-CP6108) and–IS39] [8]. Hr-HPV genotypes include HPV-16, -18, -31, -33, -35, -39, -45, -51, -52, -56, -58, -59, or -68, while low-risk HPV include HPV-6, -11, -26, -40, -42, -53, -54, -55, -61, -62, -64, -67, -69, -70, -71, -72, -73, -81, -82 subtype (IS39), -83, -84, and -89 (CP6108).

## Statistical analysis

Baseline characteristics were summarized using median and interquartile ranges for continuous variables, and proportions for categorical variables. Differences between proportions were tested by Fisher's exact test, whilst the difference between medians was computed using the Mann-Whitney U test. Log binomial logistic regression was used to determine the association between HPV infection and STI or BV infection. HPV infection was defined as follows: infection with any HPV type, infection with any low-risk type, and infection with any high-risk type. Additionally, HPV infection was defined as the detection of any types targeted by the Cervarix® vaccine (HPV-16 and -18), the detection of any HPV types targeted by the Gardasil®4 vaccine (HPV-6, -11, -16, -18), and the detection of HPV types targeted by Gardasil®9 vaccine (HPV-6, -11, -16, -18, -31, -33, -45, -52, -58) to determine the association of vaccine-preventable HPV types with STI or BV acquisition. Single HPV infection is defined as the detection of only one HPV type in the same sample. Multiple HPV infections were defined as the detection of two or more HPV types in the same sample. In cases where multiple infections were identified, individuals were counted as infected for the specific category if they have one or more infections in that category. However, these women were counted more than once when determining the prevalence of lr-HPV, and hr-HPV, if they have HPV types that belong to more than one category. A p-value less than 0.05 was considered significant. Statistical analysis was carried out using STATA v17 (STATA Corp, College Station, TX, USA) and SAS ®9.4 software (SAS Institute Inc., Cary, NC, USA).

## Results

### Baseline characteristics by HPV infection status

A total of 243 sexually active women with a median age of 23 years (interquartile range IQR 21–27 years) were included in the analysis. HPV genotypes were detected in 34% (83/243) of women (**Table 1**). The majority of women reported condom use (69.5%, 169/243) and this was similar among women with detectable HPV genotypes and those without (71.1% and 68.8%, respectively; p = 0.770). Furthermore, there were no differences in the proportion of women using hormonal contraceptives (42.5%, and 34.0%, p = 0.204), and the median number of sex partners in the past 12 months was 1 partner (IQR 1–2; p = 0.834) in both groups. More than half of the women in this cohort had intermediate BV or BV, and this did not differ significantly between women with detectable HPV genotypes and those without HPV infection (36.1% and 31.3%, respectively; p = 0.359). Nearly 15% (36/243) of women were infected with *C. trachomatis* and this was similar between groups (16.9% versus 13.8%; p = 0.569). The prevalence of *N. gonorrhoeae* (4.5%) and *T. vaginalis* (3.3%) were relatively low in this cohort and were also similar between groups (**Table 1**).

**Table 1. Sociodemographic and clinical characteristics of study participants.**

| Variable | Level | Overall (N = 243) | HPV+ (N = 83) | HPV- (N = 160) | p-value |
|---|---|---|---|---|---|
| | | % (n/N) | | | |
| Age (years) | Median (IQR) | 23 (21–27) | 22 (21–25) | 23 (21–27) | 0.234 |
| Level of education | Primary education | 0.43 (1/233) | 0 (0) | 0.64 (1/154) | 0.471 |
| | Secondary education | 72.1 (168/233) | 68.4 (54/79) | 74.0 (114/154) | |
| | Tertiary education | 27.5 (64/233) | 31.6 (25/79) | 25.3 (39/154) | |
| Condom use | Yes | 69.5 (169) | 71.1 (59) | 68.8 (110) | 0.770 |
| Frequency of condom use | Always | 4.1 (10) | 4.8 (4) | 3.8 (6) | 0.829 |
| | Sometimes | 65.4 (159) | 66.3 (55) | 65.0 (104) | |
| | Never | 30.9 (75) | 28.9 (24) | 31.3 (50) | |
| Contraceptive use | Yes | 36.9 (87/236) | 42.5 (34/80) | 34.0 (53/156) | 0.204 |
| Type of contraception | Injection | 60.2 (53/88) | 65.7 (23/35) | 56.6 (30/53) | 0.486 |
| | IUD | 1.2 (1/88) | 2.9 (1/35) | 0 | |
| | Oral | 11.4 (10/88) | 5.7 (2/35) | 15.1 (8/53) | |
| | Subdermal implant | 19.3 (17/88) | 17.1 (6/35) | 20.8 (11/53) | |
| | Condom | 8.0 (7/88) | 8.6 (3/35) | 7.6 (4/53) | |
| Number of sex partners in past 12 months | Median (IQR) | 1(1–2) | 1(1–2) | 1(1–2) | 0.834 |
| Genital examination | Abnormal | 88.2 (105/119) | 84.1 (37/44) | 90.7 (68/75) | 0.378 |
| Pelvic examination | Abnormal | 49.0 (119) | 53.0 (44) | 46.9 (75) | 0.417 |
| Candida infection | Positive | 19.3 (47) | 14.5 (12) | 21.9 (35) | 0.176 |
| BV status (Nugent score) | No BV (0–3) | 30.9 (75) | 33.7 (28) | 29.4 (47) | 0.359 |
| | Intermediate BV (4–6) | 36.2 (88) | 30.1 (25) | 39.4 (63) | |
| | BV (7–10) | 32.9 (80) | 36.1 (30) | 31.3 (50) | |
| *Chlamydia trachomatis* | Yes | 14.8 (36) | 16.9 (14) | 13.8 (22) | 0.569 |
| *Neisseria gonorrhoeae* | Yes | 4.5 (11) | 6.0 (5) | 3.8 (6) | 0.517 |
| *Trichomonas vaginalis* | Yes | 3.3 (8) | 2.4 (2) | 3.8 (6) | 0.719 |
| Any STI | Yes | 21.0 (51) | 20.5 (17) | 21.1 (34) | 1.000 |
| STI and/or BV | Yes | 71.6 (174) | 69.9 (58) | 72.5 (116) | 0.668 |
| | No | 28.4 (69) | 30.1 (25) | 27.5 (44) | |
| Co-conditions | BV only | 50.62 (123) | 49.4 (41) | 51.3 (82) | 0.759 |
| | STI only | 2.5 (6) | 3.6 (3) | 1.9 (3) | |
| | BV and STI | 18.5(45) | 16.9 (14) | 19.4 (31) | |
| | No BV or STI- | 28.4 (69) | 30.1 (25) | 27.5 (44) | |

Abbreviations: BV, bacterial vaginosis; IQR, interquartile range; IUD, Intrauterine device; STI, sexually transmitted infection. Any STI includes *Chlamydia trachomatis*, *Neisseria gonorrhoeae*, and *Trichomonas vaginalis*.

### Type-specific HPV frequency and genotype distribution

Of the 83/243 (34%) women with detectable HPV genotypes, the proportions of hr-HPV and lr-HPV infections were 54.2% (45/83) and 71% (59/83), respectively (**Fig 1A**). The most prevalent hr-HPV genotypes were HPV-51, HPV-52, HPV-45, and HPV-59 with corresponding proportions of 12% (10/83), 9.6% (8/83), 9.6% (8/83), and 9.6% (8/83), respectively. HPV-16 (8.4%, 7/83) was more prevalent than HPV-18 (1.2%, 1/83). The most frequently detected lr-HPV genotypes were HPV-81 (14.5%, 12/83), HPV-66 (9.6%, 8/83), HPV-62 (9.6%, 8/83), and HPV-11 (9.6%, 8/83). Infection with a single HPV type (54.2%, 45/83) was slightly more common than multiple HPV infections (45.8%, 38/83) in this cohort (**Fig 1B**). The multiple HPV infections were observed as follows: 28.9% of women were infected with 2 HPV types (24/83),

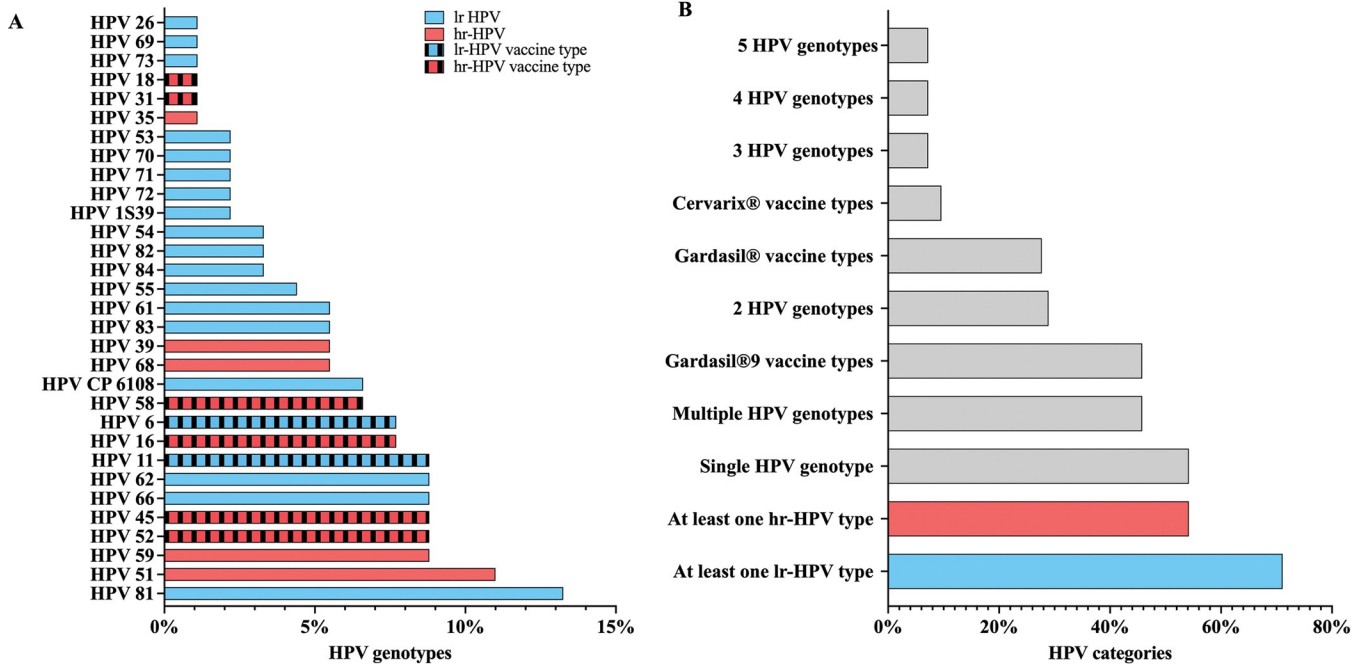

**Fig 1. A: Genotype distribution of hr- and lr-HPV among women with HPV infection (N = 83).** Frequency of HPV genotypes for all women at baseline. The salmon bars indicate hr- HPV genotypes, the blue bars indicate lr-HPV genotypes and the dotted bars indicate strains included in vaccines. **B: Type-specific HPV genotype frequency at baseline (N = 83).** The blue bars indicate lr-HPV genotypes, salmon bars indicate hr-HPV genotypes, and solid gray bars indicate, any HPV genotype, number of HPV genotypes, single and multiple HPV genotypes, and vaccine type strains. Hr-HPV is defined as infection with any of the following HPV genotypes: HPV-16, -18, -31, -33, -35, -39, -45, -51, -52, -56, -58, -59, or -68. Lr-HPV is defined as infection with any of the following HPV genotypes: HPV-6, -11, -26, -40, -42, -53, -54, -55, -61, -62, -64, -67, -69, -70, -71, -72, -73, -81, -82 subtype (IS39), -83, -84, and -89 (CP6108). Cervarix® HPV vaccine is defined as infection with HPV types-16 and -18. Gardasil®4 HPV vaccine is defined as infection with any of the following HPV types: HPV-6, -11, -16, or -18. Gardasil®9 HPV vaccine is defined as infection with HPV-6, -11, -16, -18, -31, -33, -45, -52, or -58.

7.2% with 3 HPV types (6/83), 7.2% with 4 HPV genotypes (6/83), and 7.2% with 5 HPV genotypes (6/83), respectively. Hr-HPV types targeted by the Cervarix® HPV vaccine (HPV-16 and/or 18) were detected in 9.6% (8/83) of women with HPV infection. HPV types targeted by the Gardasil®4 vaccine (HPV-6, -11, -16 and -18) were detected in 27.7% (23/83) and those targeted by Gardasil®9 vaccine types (HPV-6, -11, -16, -18, -31, -33, -45, -52 and -58) were detected in 45.8% (38/83) of women (**Fig 1B**).

## Factors associated with HPV prevalence in women

Fisher's exact tests were used to assess the relationship between HPV type and STIs and/or BV. Among women with HPV infection, infection with hr-HPV genotypes was more prevalent in women diagnosed with STIs and/or BV compared to those with no STI and/or BV (62%, 36/58 vs 36%, 9/25; p = 0.029). Gardasil®9 vaccine-type strains were more frequently detected in women diagnosed with STI and/or BV compared to women with no STI and/or BV (55.2%, 32/58 vs 24%, 6/25; p = 0.009) (**Table 2**).

Next, multivariable analysis was used to determine whether the presence of STIs and/or BV in this cohort was associated with hr-HPV or multiple HPV genotypes, as previously demonstrated [40, 41]. None of the STIs/BV were associated with any HPV or multiple HPV type infection (**Table 3**). In multivariate analysis adjusting for age, pelvic exam assessment, and number of sexual partners in the last 12 months, women diagnosed with intermediate BV or BV (aOR 2.64, 95% confidence interval (CI) 1.02–6.87, p = 0.046 and those co-infected with

**Table 2. Co-detection of HPV and other STIs and/or BV (N = 83).**

| HPV types | HPV Positive (N = 83) | | p-value |
|---|---|---|---|
| | No STI and/or BV %(n) | STI and/or BV %(n) | |
| | N = 25 | N = 58 | |
| lr-HPV | 84.0 (21) | 65.5 (38) | 0.116 |
| hr-HPV | 36.0 (9) | 62.1 (36) | 0.029 |
| Single HPV | 56.0 (14) | 53.4 (31) | 0.831 |
| Multiple HPV | 44.0 (11) | 46.6 (27) | |
| Vaccine type HPV | | | |
| Cervarix® | 4.0 (1) | 12.1 (7) | 0.425 |
| Gardasil®4 | 16.0 (4) | 32.8 (19) | 0.181 |
| Gardasil®9 | 24.0 (6) | 55.2 (32) | 0.009 |

Abbreviations: HPV, human papillomavirus; hr-HPV, high-risk human papillomavirus; lr-HPV, low-risk-human papillomavirus. P-values less than 0.05 were considered significant. One p-value was used for single and multiple HPV types.

BV and any STI (OR 3.16, 95% CI 1.15–8.64, p = 0.025) were more likely to have detectable hr-HPV genotype than women without BV (Nugent ≤3) or STI (**Table 3**).

The association between HPV infection and the exact mechanism such as markers of epithelial barrier integrity that may contribute to cervical cancer disease progression remains an active area of research. Next, we compared MMPs in women with and without HPV, and no

**Table 3. Associations between HPV, STIs, and/or BV.**

| Variable | Univariable | P-value | Multivariable | P-value |
|---|---|---|---|---|
| | OR (95% CI) | | aOR (95% CI) | |
| **Any HPV (N = 243)** | | | | |
| BV | 0.82 (0.46–1.44) | 0.486 | 0.84 (0.47–1.49) | 0.546 |
| Chlamydia | 1.27 (0.61–2.64) | 0.517 | 1.23 (0.59–2.58) | 0.584 |
| Candida | 0.60 (0.29–1.24) | 0.168 | 0.53 (0.25–1.10) | 0.089 |
| STI and/or BV | 0.88 (0.49–1.58) | 0.668 | 0.88 (0.49–1.60) | 0.685 |
| Any STI | 0.95 (0.50–1.84) | 0.889 | 0.92 (0.47–1.80) | 0.810 |
| **Multiple-type HPV infection (N = 83)** | | | | |
| BV | 1.20 (0.48–2.99) | 0.703 | 1.15 (0.44–2.98) | 0.777 |
| Chlamydia | 0.87 (0.27–2.76) | 0.810 | 0.86 (0.26–2.87) | 0.809 |
| Candida | 0.82 (0.24–2.84) | 0.757 | 0.85 (0.23–3.09) | 0.806 |
| STI and/or BV | 1.11 (0.43–2.85) | 0.831 | 1.04 (0.39–2.80) | 0.936 |
| Any STI | 0.97 (0.27–2.33) | 0.669 | 0.74 (0.24–2.26) | 0.595 |
| **Hr-HPV types (N = 83)** | | | | |
| BV | 2.50 (0.98–6.36) | 0.054 | 2.64 (1.02–6.87) | **0.046** |
| Chlamydia | 1.15 (0.36–3.68) | 0.810 | 1.17 (0.36–3.78) | 0.792 |
| Candida | 0.55 (0.16–1.91) | 0.350 | 0.57 (0.16–2.00) | 0.377 |
| STI and/or BV | 2.91 (1.10–7.70) | **0.032** | 3.16 (1.15–8.64) | **0.025** |
| Any STI | 0.94 (0.32–2.73) | 0.906 | 0.95 (0.32–2.80) | 0.930 |

CI; Confidence interval; Any STI includes *Chlamydia trachomatis*, *Neisseria gonorrhoeae*, and *Trichomonas vaginalis*. The odds ratio and corresponding p-values were determined using logistic regression analysis. aOR: adjusted Odds ratio in a multivariate model controlling for age, pelvic exam assessment, and number of sexual partners in the last 12 months. Estimates for *Trichomonas* and *Gonorrhoea* infection were excluded from the multiple and HR-HPV analysis due to the small sample size. P-values less than 0.05 were considered significant.

**Table 4. Association between HPV, STI, and/or BV and biomarkers of epithelial barrier integrity.**

| Variable | Univariable | p-value | Multivariable | p-value |
|---|---|---|---|---|
| | β (95% CI) | | β (95% CI) | |
| MMP-1 | 0.21 (-0.17–0.59) | 0.276 | 0.14 (-0.24–0.52) | 0.460 |
| MMP-2 | 0.35 (-0.04–0.74) | 0.082 | 0.30 (-0.09–0.70) | 0.133 |
| MMP-7 | 0.13 (-0.26–0.52) | 0.503 | 0.12 (-0.27–0.527) | 0.528 |
| MMP-9 | -0.12 (-0.69–0.45) | 0.672 | -0.13 (-0.69–0.44) | 0.660 |
| MMP-10 | 0.57 (0.11–1.03) | **0.016** | 0.55 (0.79–1.01) | **0.022** |

CI; Confidence interval; β-coefficients and corresponding p-values were determined using linear mixed regression analysis. A multivariable model controlled for age, pelvic exam assessment, and number of sexual partners in the last 12 months. P-values less than 0.05 were considered significant.

differences in MMP concentrations were observed (**S1 Fig**). A linear mixed regression analysis was used to estimate the association of HPV and MMPs concentrations among STI and/or BV-positive women, adjusting for age, number of sexual partners in the last 12 months, and pelvic examination assessment. Of the five MMPs tested, the concentration of MMP-10 (β = 0.55, 95% CI 0.79–1.01; p = 0.022)] was significantly increased among HPV-positive women with STI and/or BV relative to women without HPV even after adjusting for multiple comparisons. No differences were observed in other MMP concentrations of women with and without HPV (**Table 4**).

## Discussion

This study characterized the genital HPV prevalence and type-specific distribution in menstrual cup specimens of young women enrolled in a cohort study that combined an STI care model of point-of-care STI testing, immediate treatment, and expedited partner therapy. In addition, the relationship between HPV and STI and/or BV was assessed, including the relative effects on MMP concentrations. Our study demonstrated a high prevalence of hr and lr HPV infection among these women. Hr-HPV was frequently detected in STI and/or BV-positive women compared to women without. Women with STI and/or BV and with concurrent HPV exhibited significantly higher levels of MMP-10.

The overall HPV prevalence was 34% (83/243) and 54.2% of these were high-risk genotypes. This is consistent with previous studies that reported hr-HPV prevalence of 54% among women in South Africa [8, 42–44]. In contrast, other studies within the Southern Africa region have reported either lower (28–46%) [7, 45, 46] or higher (70–80%) [25, 47] hr-HPV prevalence than observed here. The inconsistency in hr-HPV prevalence within this region may be attributed to the population, sample type, collection method, and transient nature of HPV infection. Furthermore, our sample type of choice was menstrual cup pellets while others either used cervicovaginal lavage pellets [8], cervical biopsies [25], swabs or brushes [9, 48] collected from the cervix.

Studies have identified that STIs and/or BV are associated with HPV infection [17, 43], and may facilitate HPV persistence, leading to complications like cervical cancer [8]. In addition, there is evidence for multiple interactions between HPV, BV, and other STIs, and increased risk of HIV acquisition and transmission [49]. BV detection, but not STIs, showed a strong correlation with hr-HPV infection. These findings can be explained by a high burden of both BV and HPV among women of African descent. Additionally, many young, healthy, black African women have vaginal communities with low *Lactobacillus* abundance and high diversity [50, 51]. BV has been previously associated with prevalent or new HPV infections and low-grade squamous intraepithelial cervical lesions [52]. Studies have demonstrated that an

optimal female genital tract environment dominated by *Lactobacillus* species may be protective against HPV infection whereas those with diverse microbiota, reminiscent of BV, are not [53, 54]. Women with high relative abundances of *Atopobium vaginae* and *Gardnerella vaginalis* have been shown to increase the risk of acquiring new HPV infections and facilitate HPV persistence [55, 56]. It has been hypothesized that the mechanism behind increased HPV risk in women with BV may involve the disruption of epithelial barrier integrity by *G. vaginalis* [57]. These findings highlight the need for more effective treatments for BV, including *Lactobacillus*-based probiotics, which may support HPV clearance and other multi-component strategies for preventing sexually transmitted diseases (including HPV) among women.

HPV infection of the dividing basal layer requires disruption of epithelial barriers. Disruption of epithelial cells might involve the degradation of the extracellular matrix (ECM). Hr-HPV infection, including cancer progression, has been associated with increased expression of specific ECM-degrading MMPs in the cervical cancer [32, 58, 59]. In healthy tissue, MMP expression and activity is well coordinated, however, upon HPV infection this regulation is disrupted [60]. We showed that although HPV was not associated with MMP concentrations in the general population, the additive effect of concurrent HPV infection was associated with increased concentrations of MMP-10 in women with STI and/or BV. Elevated levels of MMP-10 may likely induce the expression of crucial molecules involved in angiogenesis, metastasis, and apoptosis, fostering a favourable environment that supports the survival and growth of malignant tumours [61]. Evidently, increased MMP-10 expression has been associated with resistance to apoptosis and stimulation of pro-angiogenic factors such as hypoxia-inducible factor-1 alpha (HIF)-1α and MMP-2, and pro-metastatic factors including plasminogen activator inhibitor-1 (PAI-1) and chemokine (CXC motif) receptor 2 (CXCR2) [61]. Accordingly, in vivo siRNA therapeutics targeting MMP-10 have demonstrated a reduction in tumour growth and angiogenesis [61]. In summary, the increased expression of MMP-10 in women with STIs and/or BV suggests a potential correlation with the development of cervical intraepithelial lesions. These findings collectively underscore the significance of MMPs as a viable target for therapeutic intervention in HPV-associated cervical cancer. The findings indicate that HPV alone may not be sufficient to induce the expression of MMP-10 but in combination with STIs and/or BV. The interplay between these factors could have implications for HPV persistence and the development of cervical cancer. Further research is needed to fully understand the complex interactions between cervical HPV, STIs, BV, and MMP-10 expression in women.

This study had some limitations. First, the sample size was relatively small, which impacted the statistical power and the generalizability, particularly to older women. Second, the study did not elicit whether participants were vaccinated for HPV. Although HPV vaccinations are available in the private healthcare sector, the high cost of the vaccines would have excluded this option for the majority of participants. Furthermore, the roll-out of the Cervarix® vaccine started in 2014 for 9–12-year-old girls in South Africa, while this study started in 2016 and it is therefore unlikely that participants received the vaccination in school. Nevertheless, HPV prevalence was as high or similar to reports from our region. Finally, this study only conducted a cross-sectional analysis of HPV prevalence among the cohort; hence, we were unable to assess HPV infection dynamics associated with STI and/or BV status.

In conclusion, the prevalence of hr-HPV infection among women in this population was high; underscoring an urgent need for improved cervical cancer screening and prevention programmes, including roll-out, and scale-up of HPV vaccines to curb the reported increase in HPV acquisition and cervical cancer incidence. Furthermore, the observed association between HPV infection and BV emphasizes the need to develop better BV treatments, such as biofilm dissolving agents and live biotherapeutics that could shift the vaginal microbiota

towards *Lactobacillus*-dominant communities, and multicomponent STI strategies with the potential to prevent sexually transmitted diseases and improve the reproductive health of women.

## Supporting information

**S1 Fig. Comparisons of MMP concentrations against HPV status in women at baseline (N = 243).** The t-test was used to compare median log-transformed MMP concentrations between HPV-infected and uninfected groups.
(TIF)

## Acknowledgments

We would like to thank the CAPRISA 083 study participants for their contribution to the research and the clinical and laboratory teams for the collection of clinical data and specimens.

## Author Contributions

**Conceptualization:** Wenkosi Qulu, Andile Mtshali, Nigel Garrett, Adrian Mindel, Sinaye Ngcapu, Lenine Liebenberg.

**Data curation:** Andile Mtshali, Lungelo Ntuli, Gugulethu Mzobe, Nivashnee Naicker, Lenine Liebenberg.

**Formal analysis:** Andile Mtshali, Farzana Osman, Nonsikelelo Ndlela, Gugulethu Mzobe, Lenine Liebenberg.

**Funding acquisition:** Nigel Garrett, Anne Rompalo, Adrian Mindel, Sinaye Ngcapu, Lenine Liebenberg.

**Investigation:** Wenkosi Qulu, Andile Mtshali, Lungelo Ntuli, Nigel Garrett, Anne Rompalo.

**Methodology:** Wenkosi Qulu, Andile Mtshali, Nonsikelelo Ndlela, Lungelo Ntuli, Anne Rompalo, Lenine Liebenberg.

**Project administration:** Nivashnee Naicker.

**Resources:** Nigel Garrett, Adrian Mindel.

**Supervision:** Sinaye Ngcapu.

**Validation:** Wenkosi Qulu, Andile Mtshali, Nigel Garrett.

**Visualization:** Andile Mtshali.

**Writing – original draft:** Wenkosi Qulu, Andile Mtshali, Sinaye Ngcapu, Lenine Liebenberg.

**Writing – review & editing:** Wenkosi Qulu, Andile Mtshali, Farzana Osman, Nonsikelelo Ndlela, Lungelo Ntuli, Gugulethu Mzobe, Nivashnee Naicker, Nigel Garrett, Sinaye Ngcapu, Lenine Liebenberg.

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
