## [Decision Letter · Decision Letter 0]

20 Oct 2023

PONE-D-23-24791High-risk human papillomavirus prevalence among South African women diagnosed with other STIs and BVPLOS ONE

Dear Dr. Liebenberg,

Thank you for submitting your manuscript to PLOS ONE. After careful consideration, we feel that it has merit but does not fully meet PLOS ONE’s publication criteria as it currently stands. Therefore, we invite you to submit a revised version of the manuscript that addresses the points raised during the review process.

We look forward to receiving your revised manuscript.

Kind regards,

Nontuthuzelo Iris Muriel Somdyala, Ph.D

Academic Editor

PLOS ONE

“The study was funded by the Poliomyelitis Research Foundation (PRF grant number: 19/108) and the DST-NRF Center of Excellence in HIV Prevention, which is supported by the Department of Science and Innovation, and National Research Foundation (NRF). The CAPRISA 083 cohort study was funded by a United States – South African Program for Collaborative Biomedical Research grant through the South African Medical Research Council and the National Institute of Health (AI116759). WPQ received support from the CAPRISA Research Administration and Management Training Program (Grant # G11 TW010555-01). WPQ was funded by DST-NRF CoE in HIV Prevention (grant 96354). AM is funded by DST/NRF Innovation Postdoctoral Fellowship grant number: PDG210309589501. GM is funded by DST/NRF Innovation Postdoctoral Fellowship grant number: SFP180507326699. Poliomyelitis Research Foundation (PRF grant number: 18/16).”

Reviewers' comments:

Reviewer's Responses to Questions

**Comments to the Author**

1. Is the manuscript technically sound, and do the data support the conclusions?

Reviewer #1: Yes

2. Has the statistical analysis been performed appropriately and rigorously? 

Reviewer #1: Yes

3. Have the authors made all data underlying the findings in their manuscript fully available?

Reviewer #1: Yes

4. Is the manuscript presented in an intelligible fashion and written in standard English?

Reviewer #1: Yes

5. Review Comments to the Author

Reviewer #1: The study is well conceptualized and excepted. Despite modest numbers there is valuable information about HPV, STI and bacterial vaginosis. Perhaps the only area for improvement is the reporting and discussion of the MMP analysis. A lightly more comprehensive justification of the importance of MMP will strengthen the manuscript.

6. PLOS authors have the option to publish the peer review history of their article (what does this mean?). If published, this will include your full peer review and any attached files.

Reviewer #1: No

---

## [Author Response · Author response to Decision Letter 0]

27 Oct 2023

Response to reviewers 

Reviewer #1 

1. The study is well conceptualized and accepted. Despite modest numbers, there is valuable information about HPV, STI, and bacterial vaginosis. Perhaps the only area for improvement is the reporting and discussion of the MMP analysis. A slightly more comprehensive justification of the importance of MMP will strengthen the manuscript.

Response: The text was updated to provide a more comprehensive support for the importance of investigating the contribution of HPV on mucosal barrier integrity, particularly in this context of high HIV burden. 

Line 246 – 248: The association between HPV infection and the exact mechanism such as markers of epithelial barrier integrity that may contribute to cervical cancer disease progression remains an active area of research.

Line 307-309: In healthy tissue, MMP expression and activity is well coordinated, however, upon HPV infection this regulation is disrupted [60].

Line 314-322: Evidently, increased MMP-10 expression has been associated with resistance to apoptosis and stimulation of pro-angiogenic factors such as hypoxia-inducible factor-1 alpha (HIF)-1α and MMP-2, and pro-metastatic factors including plasminogen activator inhibitor-1 (PAI-1) and chemokine (CXC motif) receptor 2 (CXCR2) [61]. Accordingly, in vivo siRNA therapeutics targeting MMP-10 have demonstrated a reduction in tumour growth and angiogenesis [61]. In summary, the increased expression of MMP-10 in women with STIs and/or BV suggests a potential correlation with the development of cervical intraepithelial lesions. These findings collectively underscore the significance of MMPs as a viable target for therapeutic intervention in HPV-associated cervical cancer.

Line: 307 – 309: In healthy tissue, MMP expression and activity is well coordinated, however, upon HPV infection this regulation is disrupted [5]. 

Line: 314 – 317: In summary, the increased expression of MMP-10 in women with STIs and/or BV denotes the invasive and metastatic characteristics of cervical cancers, emphasizing the importance of MMPs as a potential therapeutic target for intervention in HPV-associated cervical cancer. 

Additional journal requirements

Response: We have carefully reviewed the PLOS ONE style guidelines and have made the necessary adjustments to bring the manuscript into compliance with PLOS ONE style requirements. 

Response: The funders supported either the original cohort development or provided author salary/developmental support but had no involvement in this study. The text was updated to reflect this: The funders had no role in study design, data collection and analysis, decisions related to publication, or preparation of the manuscript. 

3. We note that you have indicated that data from this study are available upon request. 

Response: We comply with CAPRISA’s data-sharing policy for sensitive information and encourage interaction between requesters and study PIs. This connection is facilitated through a portal on the CAPRISA website prior to data sharing. The text is updated to reflect this: “Data will be made available through request on a dedicated portal on the CAPRISA website (https://www.caprisa.org/Pages/CAPRISAStudies)." 

Response: The ethics statement has been removed from the acknowledgment section and only appears in the method section of the manuscript

5. Please include captions for your Supporting Information files at the end of your manuscript, and update any in-text citations to match accordingly. Please review your reference list to ensure that it is complete and correct. If you have cited papers that have been retracted, please include the rationale for doing so in the manuscript text or remove these references and replace them with relevant current references.

Response: We have revised the following references to enhance the comprehensive scope of the subject and to incorporate the most up-to-date and relevant literature: 

a) We substituted (Reference 51) Di Pietro, M., et al., HPV/Chlamydia trachomatis co-infection: metagenomic analysis of cervical microbiota in asymptomatic women. New Microbiologica, 2018. 41(1): p. 513 34-41, with (Reference 8) Liebenberg, L.J., et al., HPV infection and the genital cytokine milieu in women at 395 high risk of HIV acquisition. Nature Communications, 2019. 10(1): p. 5227.

b) We replaced (Reference 55) Gabster, A., et al., High prevalence of sexually transmitted infections, and high-risk sexual behaviors among Indigenous adolescents of the Comarca Ngäbe-Buglé, Panama. Sexually Transmitted Diseases, 2019. 46(12): p. 780-787. with (Reference 53) Dareng, E.O., et al., Vaginal microbiota diversity and paucity of Lactobacillus species are associated with persistent hrHPV infection in HIV negative but not in HIV positive women. Scientific Reports, 2020. 10(1): p. 19095.

c) We replaced: (Reference 57) Castro, J., et al., Using an in-vitro biofilm model to assess the virulence potential of bacterial vaginosis or non-bacterial vaginosis Gardnerella vaginalis isolates. Scientific Reports, 2015. 5(1): p. 1-10. and (Reference 58) Masuzzo, P., et al., An end-to-end software solution for the analysis of high throughput single-cell migration data. Scientific Reports, with (Reference 51) Godoy-Vitorino, F., et al., Cervicovaginal fungi and bacteria associated with cervical intraepithelial neoplasia and high-risk human papillomavirus infections in a hispanic population. Frontiers in microbiology, 2018. 9: p. 2533. and (Reference 55) Berggrund, M., et al., Temporal changes in the vaginal microbiota in self-samples and its association with persistent HPV16 infection and CIN2+. Virology journal, 2020. 17: p. 1-9

---

## [Editor Report · Decision Letter 1]

7 Nov 2023

High-risk human papillomavirus prevalence among South African women diagnosed with other STIs and BV

PONE-D-23-24791R1

Dear Dr. Liebenberg,

We’re pleased to inform you that your manuscript has been judged scientifically suitable for publication and will be formally accepted for publication once it meets all outstanding technical requirements.

Kind regards,

Nontuthuzelo Iris Muriel Somdyala, Ph.D

Academic Editor

PLOS ONE

Additional Editor Comments (optional):

Dear Dr Liebenberg

Thank you for your submission of your research to the PLOS ONE Journal. And also, on behalf of the journal I am grateful to the review made on your manuscript to which you responded carefully as advised. After checking all comments, I am satisfied that your manuscript proceeds to publication process in which our capable team will take you through. PLOS ONE Journal values your input to the science knowledge.
---

## [Editor Report · Acceptance letter]

20 Nov 2023

PONE-D-23-24791R1 

High-risk human papillomavirus prevalence among South African women diagnosed with other STIs and BV 

Dear Dr. Liebenberg:

I'm pleased to inform you that your manuscript has been deemed suitable for publication in PLOS ONE. Congratulations! Your manuscript is now with our production department. 

Kind regards, 

on behalf of

Dr. Nontuthuzelo Iris Muriel Somdyala 

Academic Editor

PLOS ONE